# *In Vitro* Antimycobacterial Activity of Human Lactoferrin-Derived Peptide, D-hLF 1-11, against Susceptible and Drug-Resistant *Mycobacterium tuberculosis* and Its Synergistic Effect with Rifampicin

**DOI:** 10.3390/antibiotics11121785

**Published:** 2022-12-09

**Authors:** Sorasak Intorasoot, Amornrat Intorasoot, Arocha Tawteamwong, Bordin Butr-Indr, Ponrut Phunpae, Chayada Sitthidet Tharinjaroen, Usanee Wattananandkul, Sirikwan Sangboonruang, Jiaranai Khantipongse

**Affiliations:** 1Division of Clinical Microbiology, Department of Medical Technology, Faculty of Associated Medical Sciences, Chiang Mai University, Chiang Mai 50200, Thailand; 2Infectious Diseases Research Unit (IDRU), Faculty of Associated Medical Sciences, Chiang Mai University, Chiang Mai 50200, Thailand; 3Department of Microbiology, Faculty of Medicine, Chiang Mai University, Chiang Mai 50200, Thailand; 4Office of Disease Prevention and Control, 1 (ODPC 1) Chiang Mai, Department of Disease control, Ministry of Public Health Thailand, Muang District, Chiang Mai 50000, Thailand

**Keywords:** human lactoferricin 1-11, *Mycobacterium tuberculosis*, *M. tuberculosis* complex, nontuberculous mycobacteria

## Abstract

Tuberculosis is a highly contagious disease caused by the *Mycobacterium tuberculosis* complex (MTBC). Although TB is treatable, multidrug-resistant, extensively drug-resistant, and totally drug-resistant forms of *M. tuberculosis* have become a new life-threatening concern. New anti-TB drugs that are capable of curing these drug-resistant strains are urgently needed. The purpose of this study is to determine the antimycobacterial activity of D-enantiomer human lactoferricin 1-11 (D-hLF 1-11) against mycobacteria in vitro using a 3-(4,5-dimethylthiazol-2-yl)-2,5-dephenyltetrazolium bromide colorimetric assay, resazurin microplate assay, and microscopic observation drug susceptibility assay. Three previously described antimicrobial peptides, protegrin-1, AK 15-6, and melittin, with potent anti-TB activity, were included in this study. The findings suggest that D-hLF 1-11 can inhibit the growth of *M. tuberculosis* with a minimum inhibitory concentration of 100–200 µg/mL in susceptible, isoniazid (INH)-monoresistant, rifampicin (RF)-monoresistant, and MDR strains. The peptide can also inhibit some nontuberculous mycobacteria and other MTBC in similar concentrations. The antibiofilm activity of D-hLF 1-11 against the biofilm-forming *M. abscessus* was determined by crystal violet staining, and no significant difference is observed between the treated and untreated biofilm control. The checkerboard assay was subsequently carried out with *M. tuberculosis* H37Rv and the results indicate that D-hLF 1-11 displays an additive effect when combined with INH and a synergistic effect when combined with RF, with fractional inhibitory concentration indices of 0.730 and 0.312, respectively. The red blood cell hemolytic assay was initially applied for the toxicity determination of D-hLF 1-11, and negligible hemolysis (<1%) was observed, despite a concentration of up to 4 mg/mL being evaluated. Overall, D-hLF 1-11 has potential as a novel antimycobacterial agent for the future treatment of drug-sensitive and drug-resistant *M. tuberculosis* infections.

## 1. Introduction

Tuberculosis (TB) is a chronic and highly contagious disease caused by the acid-fast bacilli, *Mycobacterium tuberculosis* complex (MTBC). Among these, *M. tuberculosis* is the most widespread pathogen among humans. Although the lungs are the predominate organ of infection, the bacterium can affect all other parts of the body as well. A quarter of the world’s population is infected with *M. tuberculosis* and is at risk of acquiring TB.

Currently, the world health report estimated approximately 10 million cases of TB worldwide, with 1.2 million fatalities [1]. TB is a curable disease, while multidrug-resistant (MDR) and extensively drug-resistant (XDR) *M. tuberculosis* that are resistant to at least two potent anti-TB drugs, isoniazid (INH) and rifampicin (RF), are the new problem and have recently emerged as a significant obstacle for the success of TB treatment and control. New antimycobacterial agents for TB treatment in susceptible and drug-resistant strains are urgently needed. 

Antimicrobial peptides (AMPs) are a promising class of antimicrobial agents for the new emergence of infectious diseases including TB. In the past decade, various AMPs with potent antimycobacterial activities have been documented, including LL-37 [2] and human beta-defensin-1 to 3 [3,4]. Human lactoferricin 1-11 (hLF 1-11) is an AMPs made up of the first 11-mer obtained from the iron-binding glycoprotein lactoferrin. Throughout the decade, numerous studies demonstrated that hLF 1-11 was effective against a variety of Gram-positive bacteria, such as *Staphylococcus* spp., and Gram-negative bacteria, such as *Klebsiella* sp., *Pseudomonas* sp., and *Acinetobacter baumannii*, as well as fluconazole-resistant *Candida albicans* [5,6]. Recently, the antimicrobial activity of hLF 1-11 against *M. avium* has been investigated [7], but not every study has focused on the susceptible and drug-resistant *M. tuberculosis*. Additionally, the antibiofilm activity and the interaction between hLF 1-11 and two first-line anti-TB drugs, INH and RF, against *M. tuberculosis* has never been elucidated. 

The aim of this study is to determine the antimycobacterial activity of D-enantiomer of hLF 1-11 (D-hLF 1-11) against the susceptible and drug-resistant *M. tuberculosis* in vitro using the colorimetric MTT test, REMA, and MODS assay. Two additional MTBC, *M. bovis* and *M. microti*, and a few NTM members, *M. avium* and *M. intracellulare*, were included. The antibiofilm activity of D-hLF 1-11 toward *M. abscessus* was evaluated. Furthermore, the checkerboard experiment was used to investigate the synergistic interaction between D-hLF 1-11 and the two first-line anti-TB drugs. Finally, red-blood-cell hemolysis was carried out to determine the toxicity of the peptides in vitro.

## 2. Results

### 2.1. The Antimycobacterial Activity of L-and D-hLF 1-11 against Mycobacteria

Three available methods, including of MTT assay, REMA, and MODS assay were employed for the determination of mycobacterial growth inhibition. Based on the analysis of cell viability, the MTT colorimetric method was performed, and the results indicate that both L-and D-enantiomer of hLF 1-11 were capable of inhibiting the growth of *M. tuberculosis* H37Rv after 7 days of incubation. The rebound of the percent relative growth of *M. tuberculosis* H37Rv was observed after being treated with L-hLF 1-11, whereas D-hLF 1-11 remained active when the incubation period was extended from 7 to 14 days. The growth of *M. tuberculosis* following 14 days after being treated with L-hLF 1-11 was also observed using the REMA and MODS assay. The percent relative growth of *M. tuberculosis* H37Rv after being treated with both forms analyzed by MTT assay is shown in Table 1. The D-hLF 1-11 was provided for the following experiments.

The D-hLF 1-11 was examined with susceptible and drug-resistant strains. The results of MTT assay indicate that the peptide exhibited antimycobacterial activity in a concentration-dependent manner with a MIC of 100 µg/mL for *M. tuberculosis* H37Ra and H37Rv and 200 µg/mL for monodrug-resistant and MDR *M. tuberculosis*. Likewise, REMA was utilized for the mycobactericidal activity determination of D-hLF 1-11, and the results demonstrate that the peptide could inhibit the growth of all mycobacterial strains with the MIC of 100 µg/mL. The REMA results of the D-hLF 1-11 inhibited growth of MDR *M*. *tuberculosis* and *M. avium* are shown in Figure 1. The MODS assay was also used to assess the susceptibility of D-hLF 1-11 and the results reveal that the peptide reduced the mycobacterium growth in a manner that is compatible with the MTT assay and REMA. The demonstration of the antimycobacterial activity of D-hLF 1-11 against *M. tuberculosis* H37Rv as determined by the MODS assay is shown in Figure 2.

The antimycobacterial activity of D-hLF 1-11 was performed in some other MTBC, *M. bovis* and *M. microti*, including NTM such as *M. avium* and *M. intracellulare*. The results also indicated that the peptide could inhibit the growth of these mycobacteria. The concentration-dependent inhibition of D-hLF 1-11 against various strains of mycobacteria analyzed by the MTT assay was shown in Table 2. The MICs of D-hLF 1-11 against most mycobacteria analyzed by the MTT assay, REMA and MODS assay are summarized in Table 3.

### 2.2. The Antimycobacterial Activity of Antimicrobial Peptides against M. Tuberculosis H37Rv

In order to evaluate the performance of each test, PG-1, AK15-6, and AM-mel, which previously displayed anti-TB efficacy, were included in this study. The results indicate that these three peptides exhibit antimycobacterial activity against *M. tuberculosis* H37Rv in similar concentrations. The growth inhibition measured in term of the percent of relative growth of *M. tuberculosis* H37Rv versus the concentrations of peptides, including D-hLF 1-11, is illustrated in Figure 3.

### 2.3. Antibiofilm Activity of D-hLF 1-11 against Biofilm-Forming M. abscessus

The antibiofilm activity of D-hLF 1-11 was determined against biofilm-forming *M. abscessus* using crystal violet staining. After incubation of the biofilms with various concentrations of D-hLF 1-11, the results reveal that there is no significant difference between treated and untreated biofilm control. The percent of relative biofilm mass versus the concentrations of D-hLF 1-11 is illustrated in Figure 4.

### 2.4. The Analysis of the Synergistic Interaction of D-hLF 1-11 and Anti-TB Drugs Using the Checkerboard Assay

The checkerboard assay was performed for the synergistic analysis of the combined D-hLF 1-11 and two effective anti-TB drugs, INH and RF. FICI was calculated and the results show that the additive interaction between D-hLF 1-11 and INH and the synergistic interaction between D-hLF 1-11 and RF has FICI values of 0.730 and 0.312, respectively (Table 4). 

### 2.5. The RBC Hemolytic Assay for the Toxicity Determination of Antimicrobial Peptides

The RBC hemolytic assay was initially applied for the toxicity assessment of AMPs. It was found that less than 1% hemolysis was observed when at least 4000 µg/mL of both L- and D-hLF 1-11 was examined (Figure 5). The percentage of hemolysis gradually increased as the concentration of PG-1 and AK 15-6 increased. In contrast, complete hemolysis was observed, even after AM-mel at a low concentration was examined. The average percent of RBC hemolysis at 1000 µg/mL of L-hLF 1-11, D-hLF 1-11, PG-1, AK 15-6, and AM-mel were 0.11, 0.00, 101.07, 33.57, and 105.2, respectively.

## 3. Discussion

Tuberculosis is a major public health concern caused by *M. tuberculosis*. Although strategies to end the global TB epidemic have been launched and, hopefully, will reduce the number of TB deaths by 90% by 2030 [1], a new life-threatening problem caused by MDR, XDR, and a very-difficult-to-treat variant known as totally drug-resistant (TDR) *M. tuberculosis* is increasing annually and becoming a challenging task to control. Next generation antimycobacterial agents with low adverse effects are urgently needed.

Short AMPs are a class of naturally occurring defensive peptides that have been established as possessing antimicrobial activity against microbial pathogens with low evidence of inducing antibiotic resistance. They are also associated with the immunomodulatory functions in innate immune response [8,9]. AMPs with potent antimycobacterial activity have previously been identified, such as PR-39, a proline-arginine-rich peptide from porcine leucocytes [10], nisin A, a lantibiotic generated by *Lactococcus lactis* [11], and LLKKK18, an analogue of human cathelicidin LL-37 [12]. Recently, AMP that eliminates the MDR *M. tuberculosis* in vivo through the autophagy activation is published [13]. Hence, AMPs might be one of the most attractive agents to pave the way for future TB treatments.

Due to the fact that AMPs are short peptides, their poor stability in vivo caused by host and microbial protease enzyme cleavage becomes a major drawback and hinders their clinical utility. Currently, the modification of AMPs offers a number of advantages in terms of increased antibacterial activity, long-term stability, toxicity reduction, and bench-to-bedside application [14]. A number of techniques have been used with success, including the addition of unnatural amino acids, cyclization, N- and/or C-terminal modification, and the substitution of L- to D-enantiomer amino acids.

In this study, the antimycobacterial activity of L- and D-enantiomer of hLF 1-11 was initially investigated against *M. tuberculosis* H37Rv using MTT assay. According to the findings, both structures were almost equally effective in inhibiting the growth of mycobacteria after 7 days of incubation. Therefore, D-enantiomer remained active whether longer incubation up to 14 days was determined (Table 1). Various factors are involved in stability and behavior of peptides in vitro such as environmental temperature and pH [15,16]. Regarding mycobacterial enzymes, numerous proteolytic enzymes, such as the protease generated and released by *M. tuberculosis,* have been previously documented. Some of these enzymes are associated with host invasion, bacterial survival, and pathogenesis [17]. It has been shown that how D-hLF 1-11 remains active in prolonged incubation time may be due to its greater resistance to degradation by microbial proteases and the environment. However, the higher stability of D-hLF 1-11, compared to L-enantiomer needs to be further examined. D-enantiomer AMPs with potent antimicrobial activity and improved long-term stability have been previously established, including D-LL-37 [18], D-LAK [19], and D-Pep05 [20]. 

The antimycobacterial activity of D-hLF 1-11 was further assessed with susceptible, INH-monoresistant, RF-monoresistant, and MDR *M. tuberculosis*. Three reliable methods, including MTT assay, REMA and, MODS assay, were utilized for MIC determination. The results from all three methods are consistent and suggest that D-hLF 1-11 can inhibit the growth of these drug-sensitive and drug-resistant *M. tuberculosis* strains with an MIC of 100–200 µg/mL (Table 3). Similar to D-hLF 1-11, AMPs, such as LL-37 and defensin human neutrophil peptides that are capable of killing *M. tuberculosis* at rather high concentrations, have been documented [3,10,18,21]. More clinical isolates of mycobacteria, however, should be included for the further determination of the mycobacterial growth inhibition of D-hLF 1-11.

Three previously reported AMPs, PG-1, AK 15-6, and AM-mel, with potent antimycobacterial activity were employed as the positive control in this study. Compared to previous results, the percent of mycobacterial growth inhibition of PG-1 in this study was approximately 94.22% at 100 µg/mL, in comparison with the 96.7% at 128 µg/mL reported by Fattorini and colleagues [3]. The MIC of AK 15-6 and AM-mel in this experiment was similar to what has been previously reported [22,23]. We suggest that the slightly different MIC in this study compared to other studies may be related to the peptide purity and the techniques used for antimycobacterial activity determination. Along with *M. tuberculosis*, D-hLF 1-11 also inhibited *M. bovis* and *M. microti*, two other MTBC, and some NTM including *M. avium* and *M. intracellulare* at a similar peptide concentration. The antimicrobial activity of the hLF 1-11 against *M. avium* has previously been elucidated [7].

*M. abscessus* was utilized as a model to examine the in vitro antibiofilm activity of D-hLF 1-11, and no significant difference was observed between peptide treated and untreated control. *M. abscessus* is a rapidly growing NTM that often causing chronic pulmonary disease [24] and has been associated with biofilm formation on clinical biomaterials [25]. *Biofilm* provides several benefits to the microorganisms, such as allowing the bacteria to tolerate the antibiotics and protecting them from the host immune response [26]. Recently, it has been demonstrated that *M. tuberculosis* biofilms are more resistant to treatment with two major anti-TB drugs, INH and RF, with 50 times greater than the MIC utilized for planktonic inhibition [27]. Although D-hLF 1-11 lacked antibiofilm activity in this study, the antibiofilm properties of hLF 1-11 against yeast *Candida parapsilosis* [28] and bacterium *Pseudomonas aeruginosa* [29] have been mentioned.

The synergistic effect of D-hLF 1-11 and frontline anti-TB drugs was examined by checkerboard analysis. The additive interaction of combined peptide and INH and the synergistic effect of combined peptide and RF were obtained (Table 4). In order to check the reliability of the checkerboard assay, the MICs of INH and RF, when tested with *M. tuberculosis* H37Rv in this study, were compared to those of the previous reports, and the results indicate that the MICs of INH and RF against *M. tuberculosis* H37Rv were in the range of 0.03–0.12 µg/mL and 0.12–0.5 µg/mL, respectively [30,31]. The checkerboard result in this study highlights the benefit of combining D-hLF 1-11 with INH or/and RF for further TB treatment. The synergistic effect of combined cationic α-helical peptides and RF for mycobacterial inhibition has been previously elucidated [32].

The RBC hemolytic assay was initially applied for the toxicity determination of D-hLF 1-11 and, although a high concentration of up to 4 mg/mL was evaluated, a negligible RBC hemolysis (<1%) was observed (Figure 5). Previously, no side-effects were reported when up to 5 mg daily of a single intravenous dose was administered to healthy volunteers for 5 days [33]. For the safety assessment of antibiotic usage, the therapeutic index (TI), which is the ratio of the toxic dose to the therapeutic dose, was calculated. The TI of D-hLF 1-11 in this work was preliminary estimated to be >40. The higher the TI, the greater the safety [34]. Therefore, monitoring the efficacy of the peptide to minimize adverse effects is required before clinical trials.

Although the exact mechanism of antimycobacterial action of D-hLf 1-11 against *M. tuberculosis* is still unknown, its fungicidal action relevance to the interaction with the fungal cell membrane, has been previously described [35]. The anti-virulence activities, proteomics study, and identification of target molecule(s) of *M. tuberculosis* after being treated with D-hLF 1-11 are ongoing in our research. 

## 4. Materials and Methods

### 4.1. Bacterial Strains

Four standard strains, *M. tuberculosis* H37Ra and H37Rv, *M. bovis* BCG ATCC 35740, and *M. microti* KK 1401, are provided by Division of Clinical Microbiology, Department of Medical Technology, Faculty of Associated Medical Sciences, Chiang Mai University, Chiang Mai province, Thailand. Six clinical isolates, INH-monoresistant, RF-monoresistant, MDR *M. tuberculosis*, *M. abscessus, M. avium*, and *M. intracellularae*, are isolated from patients in northern Thailand in 2015 by the Office of Disease Prevention and Control, 1 (ODPC 1) Chiang Mai, Chiang Mai province, Thailand. The genotype the drug-resistant strains are analyzed by the commercial product, GenoType MTBDR plus assay, and is shown in Table 5. Drug-resistant phenotypes are identified using standard proportion method. The bacteria are first grown in either Lowenstein-Jensen (LJ) medium or Middlebrook 7H10 and then being subcultured into Middlebrook 7H9 broth (Becton Dickinson, Sparks, MD, USA) supplemented with 10% oleic acid, albumin, dextrose, and catalase (OADC) (Becton Dickinson, USA) or 7H9-OADC and incubated at 37 °C for 2–4 weeks. 

### 4.2. Peptides

Both L- and D-amino acids of hLF 1-11 and L-enantiomer of PG-1, AK 15-6, and AM-mel are purchased from BioBasic Inc. (Amherst, NewYork, USA) and Synpeptide Co., Ltd. (Shanghai, China). The antimycobacterial activity of antimicrobial peptides, PG-1, AK 15-6, and AM-mel, have previously been elucidated. These peptides are used as positive control in this study. The general characteristics and the previously reported MIC of each peptide against *M. tuberculosis* are summarized in Table 6. Each peptide with a purity of over 95% is dissolved in sterile phosphate buffer saline (PBS), pH 7.0, filtered using 0.2 µm syringe filter (Sartorius, Germany), aliquoted and kept at −20 °C.

### 4.3. MTT Assay

The MTT assay is carried out in accordance with earlier descriptions for antimycobacterial activity assessment of AMP [36]. In a 96-well flat-bottom plate, each peptide is two-fold serially diluted in 7H9-OADC. The drug-resistant clinical isolates and standard strains of *Mycobacterium* spp. in 7H9-OADC are adjusted with McFarland standard No. 1, diluted 1:20 in 7H9-OADC, and added (100 µL) into each well. The final concentrations of each peptide tested, including L-and D-hLF 1-11, PG-1, AK 15-6 and AM-mel, were in the range of 6.2 to 200 µg/mL. Bacterial growth control, peptide control and media control are included in each experiment. The plate is covered with lid, placed in zip-lock plastic bag and incubated at 37 °C for 7 days. Following incubation, 10 µL of 5 mg/mL MTT dye in PBS, pH 6.8 is added and the mixture is then incubated for an additional overnight. On the next day, the culture supernatant is discarded and the formazan precipitate is dissolved with 50 µL of 1:1 (*v*/*v*) of 20% sodium dodecyl sulfate and 50% N,N-dimethylformamide (SDS/DMF solution). The plate is reincubated for 3 h and the OD is measured at 570 nm using Eon microplate spectrophotometer (Biotek Instruments, Inc., Winooski, VT, USA). That the color converts from yellow to violet is indicative of the growth of mycobacteria. The MIC is defined as the lowest peptide concentration that is capable of preventing the relative growth of mycobacteria more than 99% [37]. The experiment is conducted in at least triplicate and repeated three times. To compare the stability of the hLF 1-11, both L-and D-enantiomer are incubated with the standard strain *M. tuberculosis* H37Rv at 37 °C for 7 and 14 days. After incubation, the MTT assay is performed as described above. 

### 4.4. Resazurin Microplate Assay (REMA)

The REMA is also used to evaluate the antimycobacterial activity of AMPs. The protocol is followed by the previously mentioned [38]. Briefly, two-fold serially diluted peptide in 7H9-OADC is incubated with adjusted mycobacteria (1:20) at normal atmosphere for 7 days. In a manner similar to the MTT assay, the growth control, peptide control and media control are included in each experiment. In this case, 10 μL of 0.02% resazurin (Sigma-Aldrich, Darmstadt, Germany) is added and further incubated at 37 °C for overnight. On the next day, a blue color (resazurin) becomes pink color (resorufin), which is indicative of the growth of mycobacteria. For the interpretation, the MIC is determined to be the lowest peptide concentration that devoid the change of a color from blue to pink. The experiment is conducted at least triplicate and repeated three times.

### 4.5. Microscopic Observation Drug Susceptibility (MODS) Assay

The MODS assay, a liquid culture-based technique that relies on microscopic observation of *M. tuberculosis* cording growth, is applied for antimycobacterial activity determination of MTBC. The former technique is followed, but antimicrobial peptide is used instead of the antimycobacterial drugs [39,40]. Similar to MTT and REMA, PG-1, AK 15-6, and AM-mel are included as positive control. Briefly, a total of 1 mL comprises of 900 µL of two-fold serially diluted peptide in 7H9-OADC and 100 µL of inoculum (1:20 dilution of McFarland standard No. 1) is provided in a sterile 24-well plate. The final concentrations ranged from 6.2 to 200 µg/mL. The plate is placed in zip lock plastic bag and then incubated at 37 °C for 14 days. The growth of mycobacteria is examined on day 7 and 14 under an inverted microscope at 10x magnification. The presence of the bacterial pellicles is taken as evidence of growth. When no pellicles are observed and there is no difference between the sample and the medium control well, it is categorized as negative. The MIC is defined as lowest concentration of antimicrobial peptide that give a negative result. The experiment is conducted in triplicate and repeated three times. 

### 4.6. Crystal Violet Staining

Crystal violet staining is performed for antibiofilm activity determination of D-hLF 1-11. M. abscessus that has been demonstrated to form biofilm [41], is used as a model in this study. The procedure is followed by the previously described [41,42]. Beiefly, M. abscessus colonies grown in Middlebrook 7H11-OADC are resuspended in a sterile PBS and diluted to OD_600_ 0.05 in Sauton’s minimal media. A total of 150 µL bacterial suspension is added into 96-well poly-D-lysine coated plates and further incubated at 37 °C for 4 days. After incubation, the culture medium and planktonic cells are removed and washed once with PBS. In this case, 150 μL of serial two-fold diluted peptide in the range of 6.2–200 µg/mL are added into a 4-day old biofilm and further incubated at 37 °C for 24 h. The peptides are removed and washed once with 150 µL PBS. Next, 100 µL of 0.05% crystal violet solution is added in each well and incubated at room temperature (RT) for 30 min. The dye solution is discarded and microwells are washed once with PBS and air-dried. The crytal violet is extracted with 300 µL of 30% acetic acid and further incubated at RT for 30 min. Finally, the absorbance is measured at 562 nm using Eon microplate spectrophotometer (BioTek Instruments, Inc., Winooski, VT, USA). The percent of relative biofilm mass is calculated [43]. Untreated wells are defined as 100% biofilm formation. The experiment is conducted in duplicate and repeated three times. 

### 4.7. Checkerboard Assay

Checkerboard assay is applied for investigation of the synergistic effect of D-hLF 1-11 in conjunction with anti-TB drug, INH or RF. The procedure is followed by the aforementioned [44,45]. In 96-well plate, the D-hLF 1-11 is two-fold serially diluted along the ordinate, while the anti-TB drug is diluted along the abscissa. In this case, 50 μL of two-fold serially diluted AMP in 7H9-OADC is mixed with 50 µL of either INH or RF. Next, 100 μL of approximately 10^5^ CFU/mL of adjusted *M. tuberculosis* H37Rv is added. The final concentration of D-hLF 1-11 is ranged from 3.12–200 µg/mL. The final concentrations of INH and RF are 0.001–0.8 µg/mL and 0.002–2 µg/mL, respectively. Plate is incubated at normal condition for 7 days. The REMA is utilized for MIC determination. The experiment is conducted in triplicate and repeated three times. The interpretation of D-hLF 1-11 and anti-TB drug interaction is performed according to the prior publication [46]. The fractional inhibitory concentration (FIC) is defined as the lowest concentration of peptide and the anti-TB drug in combination that can inhibit the growth of *M. tuberculosis* divided by the MIC of either peptide or anti-TB drug. The synergistic effect between peptide and anti-TB drug is denoted by the fractional inhibitory concentration index (FICI) and interpreted by following this criteria: FICI < 0.5, synergistic effects; 0.5 < FICI < 1, additive effects; 1 < FICI < 4, no interactions; FICI > 4, antagonistic effects. The FICI is calculated by the following formula:FICI = FIC_peptide_ + FIC_anti-TB drug_ = [peptide]/MIC_peptide_ + [anti-TB drug]/MIC_anti-TB drug_

### 4.8. Red Blood Cells Hemolytic Assay

In order to determine the toxicity of antimicrobial peptides, an in vitro red blood cell hemolytic experiment is carried out. The protocol is followed exactly as it has been reported earlier [32]. Three milliliter of human whole blood group O is collected in EDTA tube and centrifuged at 450 g for 10 min. Plasma is discarded and RBC pellet is washed thrice with 10 mL sterile PBS, pH 7.0. The final peptide concentration is in the range of 15.6–1000 µg/mL and up to 4000 µg/mL for both L- and D-hLF 1-11. The peptide suspension buffer (PBS) and 1% triton X-100 are included as 0% hemolysis and 100% hemolysis, respectively. Finally, plate is centrifuged at 4000× *g* for 5 min, room temperature (Eppendorf Centrifuge 5804, Hamburg, Germany). In this case, 100 μL of supernatant is transferred into new 96-well plate and the optical density (OD) is measured at 576 nm using Eon microplate spectrophotometer (BioTek Instruments, Inc., Winooski, VT, USA). The following equation below is used to calculate the percentage of RBC hemolysis: 

Here, % hemolysis = (absorbance of peptide-absorbance of peptide suspension buffer)/(absorbance of 1% triton X-100-absorbance of peptide suspension buffer) × 100

## 5. Conclusions

In conclusion, in vitro antimycobacterial activity of D-hLF 1-11 is determined using MTT assay, REMA, and MODS assay. The peptide is effective for inhibiting of the growth of susceptible, drugs-resistant *M. tuberculosis*, and NTM with the MIC of approximately 100–200 µg/mL. Therefore, more clinical isolates of mycobacteria need to be included for the further determination of the mycobacterial growth inhibition of D-hLF 1-11. The antibiofilm activity of D-hLF 1-11 is examined against biofilm-forming *M. abscessus*. However, no biofilm inhibition is observed. The peptide prefers the additive effect with INH and synergistic effect with RF for the growth inhibition of *M. tuberculosis*. Negligible hemolysis is obtained while a high concentration of up to 4 mg/mL peptide is evaluated. Overall, D-hLF 1-11 is highly stable and safe, and would be warranted as a novel antimycobacterial agent for clinical exploration in TB and drug-resistant TB treatment in the future. 

## Figures and Tables

**Figure 1 antibiotics-11-01785-f001:**
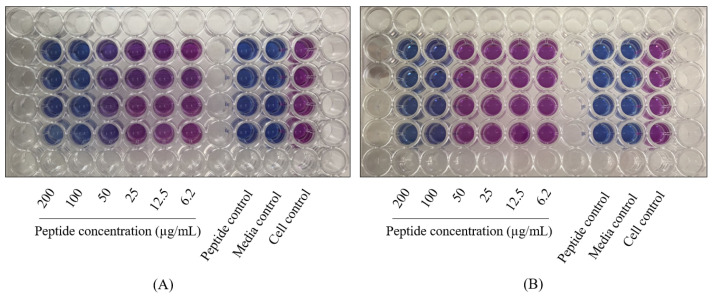
The mycobacterial growth inhibition of D-hLF 1-11 against MDR *M. tuberculosis* and *M. avium* analyzed by REMA test. The MDR *M. tuberculosis* (**A**) and *M. avium* (**B**) are incubated with various concentrations of D-hLF 1-11 ranging from 6.2–200 µg/mL and analyzed by REMA. The MIC is defined as the lowest peptide concentration that devoid the change of a color from blue to pink.

**Figure 2 antibiotics-11-01785-f002:**
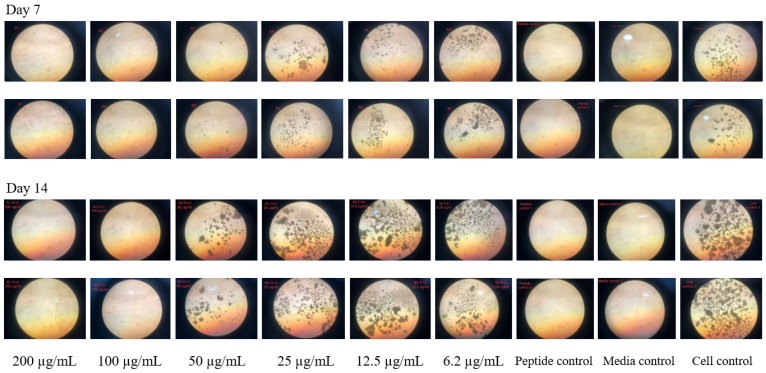
The determination of antimycobacterial activity of D-hLF 1-11 against *M. tuberculosis* H37Rv using MODS assay. *M. tuberculosis* H37Rv is incubated with various concentrations of D-hLF 1-11 ranging from 6.2–200 µg/mL. After incubation, the cording growth of *M. tuberculosis* is observed under the inverted microscope. The MIC is defined to be the lowest concentration of the peptide that give a negative result (no pellicles are observed).

**Figure 3 antibiotics-11-01785-f003:**
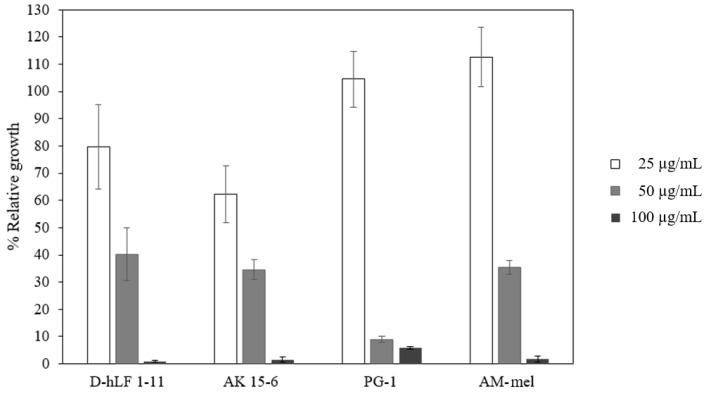
The mycobacterial growth inhibition of antimicrobial peptides against *M. tuberculosis* H37Rv measured by MTT assay. The y-axis is the percent relative growth whereas the x-axis is the tested peptides ranging from 25 (open bar), 50 (gray bar), and 100 (dark-gray bar) µg/mL. The means ± standard deviation (SD) from three independent experiments are indicated.

**Figure 4 antibiotics-11-01785-f004:**
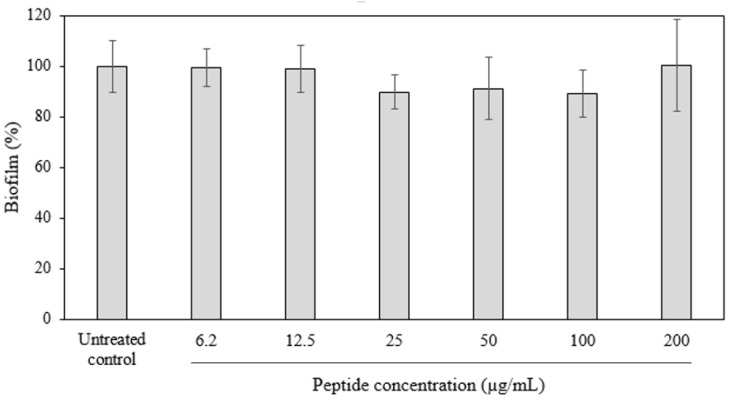
The antibiofilm activity of D-hLF 1-11 against the biofilm-forming *M. abscessus*. The antibiofilm activity of D-hLF 1-11 is determined using crystal violet staining. The 4-day old of *M. abscessus* biofilms are incubated with various concentrations of peptide ranging from 6.2–200 µg/mL. The means ± standard deviation (SD) from three independent experiments are indicated. No significant difference is observed between treated and untreated control.

**Figure 5 antibiotics-11-01785-f005:**
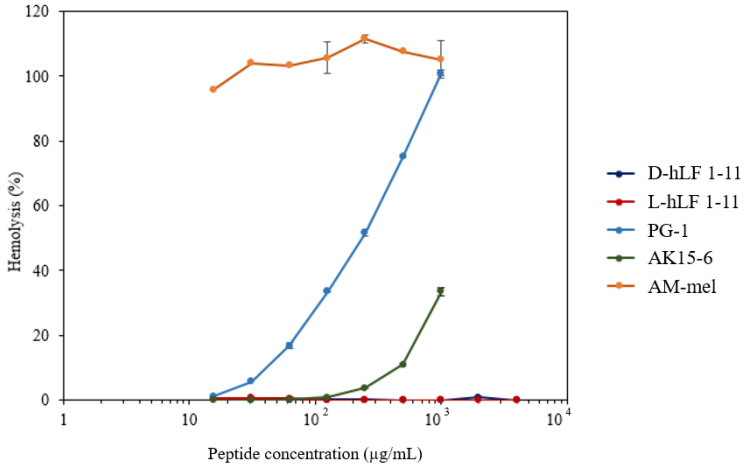
The red blood cell hemolytic assay for toxicity determination of AMPs. The percent hemolysis versus the log concentrations of antimicrobial peptides is plotted. The concentration of PG-1 (sky blue), AK 15-6 (green), and AM-mel (orange), ranging from 15.6–1000 µg/mL and up to 4000 µg/mL for both L- (red) and D-enantiomer (blue) of hLF 1-11 are examined. Means ± standard deviation (SD) in triplicates are shown.

**Table 1 antibiotics-11-01785-t001:** The antimycobacterial activity of L-and D-hLF 1-11 against *M. tuberculosis* H37Rv analyzed by MTT assay.

Peptide Concentration (µg/mL)	Mean of Relative Growth ± SD (%)			
	7 Days of Incubation		14 Days of Incubation	
	L-hLF 1-11	D-hLF 1-11	L-hLF 1-11	D-hLF 1-11
0 *	100.00 ± 5.92	100.00 ± 5.92	100.00 ± 5.35	100.00 ± 5.35
6.2	83.51 ± 3.66	95.62 ± 13.39	112.52 ± 3.40	96.89 ± 1.40
12.5	67.66 ± 5.21	64.11 ± 7.47	101.49 ± 3.64	102.70 ± 0.97
25	46.23 ± 14.93	57.55 ± 14.81	92.49 ± 5.42	72.09 ± 1.84
50	43.40 ± 9.82	41.71 ± 15.71	107.15 ± 4.61	67.57 ± 11.59
100	11.77 ± 2.33	2.09 ± 0.19	64.20 ± 6.16	31.25 ± 0.21
200	5.18 ± 0.66	0.97 ± 0.19	99.81 ± 5.68	0.56 ± 0.05

* Bacterial cell control.

**Table 2 antibiotics-11-01785-t002:** The concentration-dependent inhibition of D-hLF 1-11 against various strains of mycobacteria analyzed by the MTT assay.

Peptide Concentration (µg/mL)	Mean of Relative Growth ± SD (%)								
	*M. tuberculosis* H37Ra	*M. tuberculosis* H37Rv	INH-Resistant *M. tuberculosis*	RF-Resistant *M. tuberculosis*	MDR *M. tuberculosis*	*M. bovis*	*M. microti*	*M. avium*	*M. intracellulare*
0 *	100.00 ± 0.54	100.00 ± 17.34	100.00 ± 11.69	100.00 ± 7.02	100.00 ± 15.04	100.00 ± 4.32	100.00 ± 3.34	100.00 ± 3.91	100.00 ± 6.74
6.2	78.32 ± 6.20	119.92 ± 6.75	114.64 ± 3.84	165.75 ± 21.00	93.90 ± 10.40	104.63 ± 6.09	116.24 ± 4.13	104.73 ± 2.63	112.24 ± 7.25
12.5	83.19 ± 5.38	97.32 ± 23.52	114.03 ± 7.87	94.93 ± 20.70	113.35 ± 16.62	113.62 ± 2.35	95.12 ± 10.80	108.50 ± 3.66	117.28 ± 8.14
25	82.52 ± 8.52	86.26 ± 19.18	70.55 ± 16.30	133.33 ± 6.41	73.16 ± 12.24	118.13 ± 5.34	84.73 ± 10.26	95.52 ± 9.42	122.60 ± 6.05
50	64.36 ± 3.42	37.78 ± 15.62	53.58 ± 17.38	130.72 ± 11.90	44.93 ± 8.48	90.07 ± 5.96	34.51 ± 5.72	90.72 ± 6.53	84.16 ± 4.58
100	0.68 ± 0.96	0.91 ± 0.23	4.96 ± 0.07	9.81 ± 4.67	5.51 ± 3.00	43.25 ± 0.98	0.72 ± 0.21	13.58 ± 0.62	9.47 ± 1.73
200	0.46 ± 0.79	0.36 ± 0.00	0.21 ± 0.05	0.51 ± 0.21	0.47 ± 0.27	16.56 ± 0.44	0.64 ± 0.00	2.82 ± 0.71	5.37 ± 0.86

* Bacterial cell control.

**Table 3 antibiotics-11-01785-t003:** The MICs of D-hLF 1-11 antimicrobial peptide against susceptible and drug resistant *M. tuberculosis* analyzed by the MTT, REMA and MODS assay.

Strains of Mycobacteria	MIC (µg/mL)		
	MTT Assay	REMA	MODS Assay
*M. tuberculosis* H37Ra	100	100	100
*M. tuberculosis* H37Rv	100	100	100
INH-monoresistant *M. tuberculosis*	200	100	200
RF-monoresistant *M. tuberculosis*	200	100	200
MDR *M. tuberculosis*	200	100	200
*M. bovis*	>200	100	>200
*M. microti*	100	100	200
*M. avium*	>200	100	ND *
*M. intracellulare*	>200	100	ND

* ND = not determined.

**Table 4 antibiotics-11-01785-t004:** The checkerboard assay for antimycobacterial activity determination of the combined anti-TB drugs and D-hLF 1-11.

Drug-Peptide Combination	MIC (µg/mL)		FIC	FICI
	Alone	Combined		
INH	0.025	0.012	0.48	0.730
D-hLF 1-11	100	25	0.25	
RF D-hLF 1-11	0.5 100	0.031 25	0.062 0.25	0.312

**Table 5 antibiotics-11-01785-t005:** Mutations identification of *M. tuberculosis* mutants used in this study.

Strain	Phenotype	Mutation Identification		
		*rpo*B	*kat*G	*inh*A
Isoniazid monoresistant *M. tuberculosis*	INH-resistant	-		C-15T
Rifampicin monoresistant *M. tuberculosis*	RF-resistant	H526D	-	-
MDR *M. tuberculosis*	INH- & RF-resistant	D516V	S315T	-

**Table 6 antibiotics-11-01785-t006:** The general characteristics and the previously reported MIC of each antimicrobial peptide used in this study.

Antimicrobial Peptide	Amino Acid Sequence	Length	Source	MIC (µg/mL)	Test	Reference
PG-1	RGGRLCYCRR RFCVCVGR	18	*Sus domesticus*	128	Plate count assay	[3]
AK 15-6	AVKKLLRWWS RWWKK	15	Mycobac teriophage	37.5	Kinetic killing assay	[22]
AM-mel	GIGAVLKVLT TGLPALISWI KRKRQQ	26	*Apis mellifera*	32–64	REMA	[23]
hLF 1-11 (D & L-form)	GRRRRSVQWC A	11	*Homo sapiens*	-	-	This study

## Data Availability

All data related to this study can be found in the main manuscript.

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
