# Peer review of "In Vitro Antimycobacterial Activity of Human Lactoferrin-Derived Peptide, D-hLF 1-11, against Susceptible and Drug-Resistant Mycobacterium tuberculosis and Its Synergistic Effect with Rifampicin"

_antibiotics, 2022, doi:10.3390/antibiotics11121785_

Round 1

Reviewer 1 Report

Comments to the authors

The main objective of this study was to investigate the in vitro activity of human lactoferrin-derived peptide against susceptible and drug-resistant Mycobacterium tuberculosis and its synergistic interaction among the peptide and the two first line anti-TB antibiotics, isoniazid and rifampicin.

The work could be interesting, but the manuscript needs to be clarified or improved. I have some comments and suggestions for the authors

First the all, the manuscript structure does not follow the guidelines of the Antibiotics journal (see, https://www.mdpi.com/journal/antibiotics/instructions). The Result and Discussion Section have to appear before the M&M Section.

Introduction Section:

Line 60-67. The authors talk about the possibility of the degradation of the study peptide and other LL-37, however, this degradation has not studied. All the paragraph should be removed from the Introduction Section and discussed in the Discussion Section as a potential limitation.

Material and Methods Section:

How many replicates have been made of each experiment (assay)? duplicate, triplicate…Please, clarified this point. 

Line 84, Bacterial strains. Eight strains were studied among them Mycobacterium tuberculosis H37Rv, however in Figure 6 legend, there is Mycobacterium tuberculosis H37Ra.

Line 85-87, I suppose that the strains INH-resistant, RIF-resistant and MDR M.tuberculosis are clinical isolates, doesn't it?  and M.avium and M.intracellulare? Please, clarified this point.

Line 112, …” the final peptide concentration…” which peptide is? I suppose that is hLF 1-11 but what form is, D or L or both? The sentence should be rewritten.

Line 133-134, This sentence and Figure 1 should be moved to Result Section as an example of the REMA assay result.

Line 136, ….” the antimicrobial peptide was tested against various strains of M.tuberculosis….” What strains? clinical isolates? This point should be clarified.

Line 152-153, This sentence and Figure 2 should be moved to Result Section as an example of the MODS assay result.

Results Section:

Line 194-195, … “results indicated that both L-and D-enantiomer of hLF 1-11…”  See comment in M&M Section. It is important to explain that both form have been tested and in what strains, only in M.tuberculosis H37Rv? if yes, what happen with the others strains?

Figure 3. The result could be explaining in the text and the Figure should be removed. In addition, this extend period to incubation should be comment in the M&M Section, 2.3 MTT assay.

Line 197-198, …” the three previously reported peptides…” …” were included as positive control in this study. It is necessary to clarify the role of the PG-1, AK15-6 and AM-mel peptides in the M&M section, explaining in what method have been used and in what concentration.

There are many figures, it would be better modified the values in Figures 5, 6 and 7 in a Table.

Discussion Section:

This Section should be shorted and rewritten. The first paragraph is an introduction and should be removed. The second one is a mix between introduction and discussion.

The authors should be considered that their results include a very little strains (and only three clinical isolates). This is an important limitation. A comment about this point should be included.

Reviewer 2 Report

In this study, the colorimetric MTT test, REMA and MODS assay was used to investigate the in vitro antimycobacterial activity of D-enantiomer of hLF 1-11 (D-hLF 1-11)  against M. tuberculosis H37Rv, INH-monoresistant, RF-monoresistant and MDR M. tuberculosis. 

- You need to clearly state the aim of your study at the end of introduction.

- Why you selected other AMPs from other studies to test and why you selected those specifically.

- Change the title of table 1 to : mutants used in this study.

- Why do not you track growth using absorbance rather than using MTT and REMA.

- In vitro and in vivo should be in italic.

- You need to write the result of each experiment separately and provide subtitles. 

- the way you put the results is very confusing.

- you need to see the effect of these peptides on biofilm formation and virulence factors. more experiments need to be done.

Reviewer 3 Report

Title:

 In vitro antimycobacterial activity of human lactoferrin-derived 2 peptide, D-hLF 1-11, against susceptible and drug-resistant Mycobacterium tuberculosis and its synergistic effect with rifampicin

1.      Background is quite broad and hence it is hard to understand why is needed of latoferricin to test against TB.

2.      There are too many abbreviations used in the abstract.

3.      Justification for this study is poorly explained in the introduction part.

4.      Improve the introduction, need it to make it simple but ensure technically sound and understandable.

5.      Many abbreviations are not written in full before it appears subsequently. Please ensure.

6.      Work is more focused on tuberculosis, but least is justified the gap addressed in this research through literature with reference to M. tuberculosis.

7.      What they really want to achieve with AMP with M. tuberculosis is not concentrated.

8.      Please write strong background and link it to justify the need for this study with a clear understanding of what is main theory or mechanism or process you are going to reveal. Just mentioning what tests are being performed or briefly explaining of methods applied does not make good sense to justify this work's objectives. After all, clearly write your objectives for the research.

9.      Species names must be abbreviated.

10.  L113-μlg/mL, is it μl/mL or μg/mL?

11.  Page 128-In a mammer similar? Please check such corrections throughout the manuscript.

12.  In figure 2 peptide control is missing.

13.  In methodology, mention the statistical analysis. In Figures 3 and 4 statistical analysis is required.

14. Figures 6 and 7: legends should be within the figure's margins.

15.  They have used the literature-cited drugs in experimentation. They should include commercial drugs to compare the outcome with selected lactoferricin treatment.

16.  In discussion please refer to the relevant table or figure please.

17. In Tables 3 and 4, the authors are ensured to state or compare commercial peptides with other studies and this work.

Round 2

Reviewer 1 Report

I agree with the manuscript in the present form

Reviewer 2 Report

Thank you so much for addressing my comments. However, you still need to test the anti-biofilm activity of hLF 1- 11 against mycobacteria not Candida parasilosis 

Reviewer 3 Report

The authors need to improve the conclusion. The conclusion is too short. Better it would be if they concluded in the conclusion in a quantifiable way. They clearly conclude from each parameter, with some statement of limitation of in their research works and recommendations for future work. 
